

# New record of podocopid ostracods from Cretaceous amber

He Wang[1], Mario Schädel[2], Benjamin Sames[3] and David J. Horne[4]

[1] State Key Laboratory of Palaeobiology and Stratigraphy, Nanjing Institute of Geology and Palaeontology and Center for Excellence in Life and Paleoenvironment, Chinese Academy of Sciences, Nanjing, China
[2] Zoomorphology group, Department of Biology II, Ludwig-Maximilians-Universität München, Planegg-Martinsried, Germany
[3] Department of Geodynamics and Sedimentology, University of Vienna, Vienna, Austria
[4] School of Geography, Queen Mary University of London, London, United Kingdom

## ABSTRACT

Burmese Cretaceous amber (∼99 Ma, Myanmar) is famous for the preservation of a wide range of fauna and flora, including representatives of marine, freshwater and terrestrial groups. Here, we report on three ostracod specimens, that came visible as syninclusions to an aquatic isopod. The three specimens represent three different taxa, that were found preserved in a single piece of amber. One of the described specimens was studied using μCT scanning data. On the basis of general carapace morphology we assign all three to the group Podocopida, and (tentatively) its ingroup Cypridocopina. A lack of visibility of more particular diagnostic features such as adductor muscle scars and details of the marginal zone precludes a further identification, but we discuss possible affinities with either the marine-brackish group Pontocypridoidea or the non-marine group Cypridoidea. The taphonomy indicates that the studied ostracods had been subject to limited (if any) post-mortem transport, which could be consistent with marginal marine environments.

Corresponding author
He Wang, hwang@nigpas.ac.cn

## INTRODUCTION

Ostracods (also known as "seed shrimps") are small crustaceans with a bivalved carapace (e.g., *Horne, Cohen & Martens, 2002*). Adult ostracods are typically 0.5–2 mm long, but can also be smaller or much larger, for example the marine *Gigantocypris* can be more than 30 mm (*Poulsen, 1962*; *Horne, Cohen & Martens, 2002*; *Brusca & Brusca, 2003*). Ostracods are the most commonly preserved fossil arthropods, which can be dated back to Early Ordovician (*Horne, 2005*). The earliest ostracods are all marine and the first undoubted non-marine representatives of the group are of Early Carboniferous age (*Rodriguez-Lazaro & Ruiz-Muñoz, 2012*). At the present day, non-marine ostracods can be found in most non-marine aquatic ecosystems including freshwater and saline lakes, streams and rivers, springs, wetlands, temporary ponds, and groundwater (*Horne, Meisch & Martens, 2019*). They can even be found in semi-terrestrial habitats such as moist soils with leaf litter (*Rodriguez-Lazaro & Ruiz-Muñoz, 2012*). However, most of the relatively few records of ostracods preserved in amber (e.g., *Keyser & Weitschat, 2005*; *Keyser & Friedrich, 2017*;

*Matzke-Karasz et al., 2019*) are from the Cenozoic and, although the number of kinds of organisms trapped in Burmese Cretaceous amber from Myanmar has increased exponentially over the past few years (including flowers, fungi, scorpions, spiders, crabs, frogs, dinosaurs and insects; *Ross, 2019*), only one ostracod (a marine myodocopan) has so far been reported (*Xing et al., 2018*). Here, we report podocopan ostracods from amber of Myanmar ("Burmite"), the age of which has been biostratigraphically constrained to be late Albian–early Cenomanian (latest Early to earliest Late Cretaceous; *Cruickshank & Ko, 2003*). *Shi et al. (2012)* further constrained the age geochronologically based on U-Pb zircon ages from the volcanoclastic rock matrix, containing the amber, to lowermost Cenomanian, ~99 million years. In this study we describe three fossil remains of Ostracoda in Burmese amber along with a careful interpretation about their systematic position and critically discuss their values as palaeoenvironmental indicators.

## MATERIALS & METHODS

A single piece of amber (26.4 mm feret diameter) is the focus of this study. It was commercially obtained by Mark Pankowski (Rockville, Maryland, USA) and was donated to the collection of the Natural History Museum Vienna ("Naturhistorisches Museum Wien", NHMW) in April 2017 (Thomas Nichterl, collection manager at NHMW, pers. comm., 2020). The amber piece is available under the collection number NHMW-2017/0052/0001. The amber piece comes from one of the commercial mining sites in the Hukawng Valley in the Kachin Province of Myanmar (*Haug et al., 2020*). Due to being commercially acquired, stating a more precise provenance is not possible.

Microscopic images were made using a Keyence VHX-6000 digital microscope. The focus-stacking function of the digital microscope was used to create in-focus images of three-dimensional objects despite the limitation of the depth-of-field. To gather X-ray micro-computer tomography (CT) data, a Baker Hughes (General Electrics) "phoenix nanotom m" computer tomograph was used along with the acquisition software "datos|x". The CT imaging was performed at the zoological State collection in Munich. A current of 100 kV was used to scan the object. The amber piece was rotated 360 degrees in 1440 steps. The final reconstruction of the volume data was done using VGStudio MAX 2.2.6.80630 (Volume Graphics). The achieved voxel size of the volume was 2.81295 μm. Drishti 2.6.4 (GNU) was used for volume rendering. In some cases, more than one transfer function was applied to show structures with different X-ray qualities. Regular (two-dimensional) images and red-cyan stereo anaglyphs were exported. GIMP 2.10 (GNU) was used to optimize the histogram, and enhance colour, brightness and contrast of the final images. Inkscape (versions 0.92.3 and 0.92.4, GNU) was used to create the figure plates. Measurements were performed using ImageJ (FIJI, public domain).

### Systematic palaeontology

The higher classification draws mainly on schemes published by *Horne (2002)*, *Hou, Gou & Chen (2002)* and *Smith et al. (2015)*.

Ostracoda *Latreille, 1806* (= Ostrachoda *Latreille, 1802*)
Podocopa *Sars, 1866*
Podocopida *Sars, 1866*
Cypridocopina *Jones, 1901*

All three ostracod taxa are assigned to Cypridocopina on the basis of their overall carapace morphology, including rounded subtriangular outline in lateral view, compressed fusiform outline in dorsal/ventral view, and smooth or lightly pitted external surface. Neither with light microscopy nor with computer-tomography have we been able to resolve any diagnostic features, such as adductor muscle scars or internal details of the marginal zone, that might lead to a more precise identification.

Taxon A

Fig. 1A; Fig. 2; Suppl. 1

Material. One articulated carapace, moderately well preserved.

Dimensions. L: 0.57 mm, H: 0.34 mm, W: 0.17 mm.

Locality and horizon. Hukawng Valley, Kachin Province, Myanmar (E96°36′15″, N26°13′47″, accuracy of about 10 km); lowermost Cenomanian, lowermost Upper Cretaceous.

Description. Carapace small, elongate subtriangular in lateral view; slender, fusiform (spindle-shaped) with bluntly rounded extremities in dorsal view. Left valve slightly larger than right valve and overlapping right valve along all margins but most markedly on the ventral margin and at the highest point of the dorsal margin. Left valve possibly with alveolus behind a rostrum. Maximum height at one third of length from anterior margin. Anterior cardinal distinct, obtuse-angled (approx. 140°), forming the highest point. Posterior cardinal angle weakly rounded and obtuse with about 160°. Anterior margin broad and slightly infracurvate, almost equicurvate with a moderately long, nearly straight dorsal part. Posterior margin distinctly narrower than anterior one, rounded and equicurvate, having a short slightly curved dorsal part. Dorsal margin slightly convex and inclined towards posterior end, about 20°. Ventral margin almost straight, slightly concave at mid-length. Surface smooth. Internal features not observable.

Remarks: The left valve shows shallow indentation near the anterior end of the ventral margin that could be interpreted as an alveolus behind a rostrum, i.e., a ''beak'' such as is diagnostic of the group *Cypridea* (Superfamily Cypridoidea) (Fig. 2H). However, there is no trace of its equivalent in the right valve, and the feature may actually represent damage to the valve. Based on the gross morphological similarities, Taxon A belongs to either Pontocypridoidea or Cypridoidea. However, the strong asymmetry of the right and left valves perhaps favours an assignment to the group Cypridoidea but this is not conclusive.

Taxon B

Fig. 1B

Material. One articulated carapace, well-preserved.

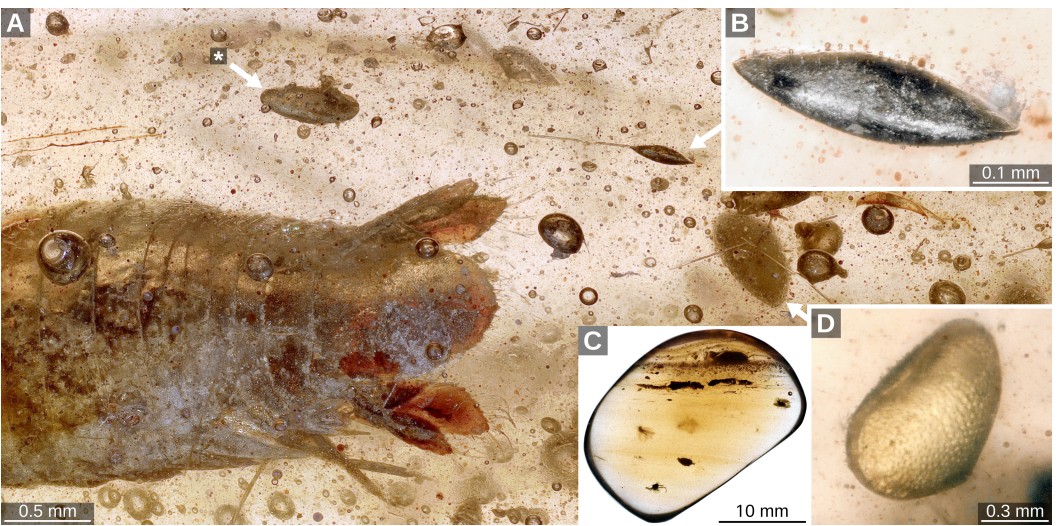

**Figure 1** (A) Overview image depicting the relative position of the ostracods (white arrows); (B) Taxon B with numerous vesicles at or right above the lateral pore canals; (C) overview image of the amber piece. (D) Taxon C; asterisk: Taxon A.

Locality and horizon. Hukawng Valley, Kachin Province, Myanmar (E96°36′15″, N26°13′47″, accuracy of about 10 km); lowermost Cenomanian, lowermost Upper Cretaceous.

Description. Carapace small and fusiform (spindle-shaped) with sharp extremities in dorsal view; surface smooth. Valves approximately equal in size, without evident overlap. Internal features not observable.

Remarks: We have been unable to obtain a clear lateral view of this carapace, which appears to differ from those of taxa A and C by having sharper anterior and posterior extremities and no obvious overlap of the valves. We speculate that Taxon B belongs to either Pontocypridoidea or Cypridoidea according to preserved features.

Taxon C

Fig. 1C

Material. One articulated carapace, well-preserved.

Locality and horizon. Hukawng Valley, Kachin Province, Myanmar (E96°36′15″, N26°13′47″, accuracy of about 10 km); lowermost Cenomanian, lowermost Upper Cretaceous.

Description. Carapace small, rounded subtriangular in lateral view. Left valve slightly overlapping right valve along ventral and posterior margins. Maximum height in front of mid-length. Anterior margin broad, almost equicurvate. Posterior margin narrower than anterior one, rounded and equicurvate. Carapace distinctly punctate tending to reticulation. Internal features not observable.

Remarks: The carapace is similar in lateral outline, but laterally more inflated, than that of Taxon A. Moreover, Taxon A has an apparently smooth exterior while Taxon C is distinctly

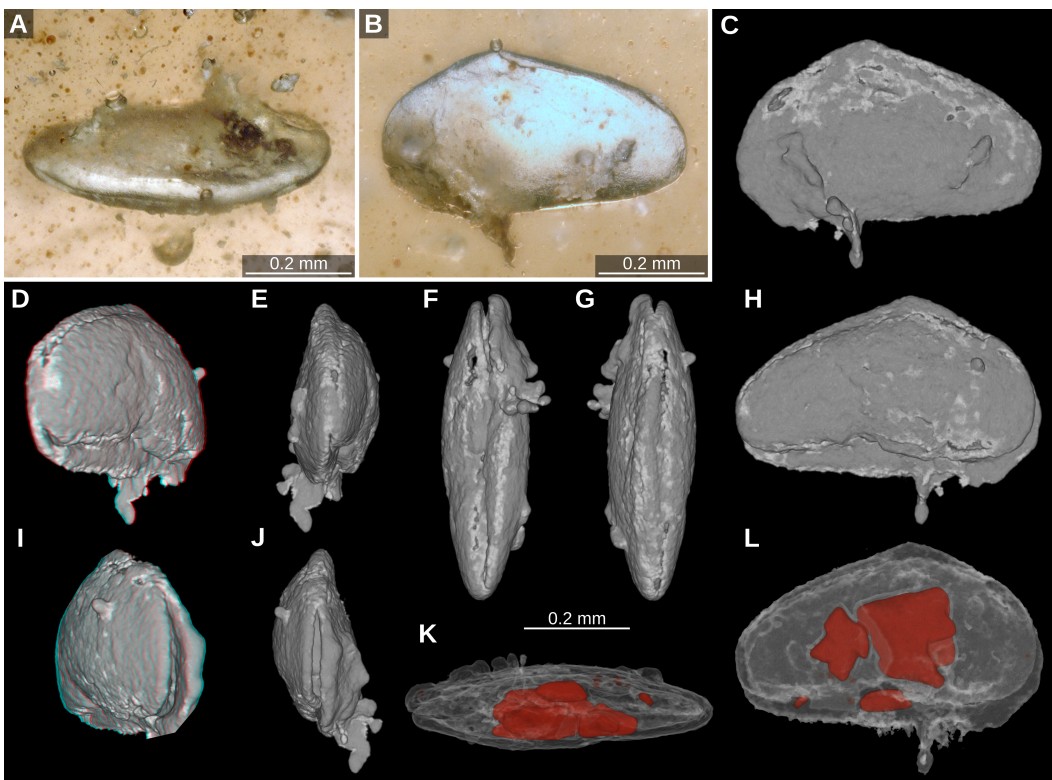

**Figure 2** Taxon A. (A–B) light microscopy, mixed translucent and reflected light. (A) dorsolateral view. (B) left lateral view, side 1; (C–L) volume rendering images based on μCT data. (C) left lateral view; (D) red-cyan stereo anaglyph, posterolateral view; (E) posterior view; (F) ventral view; (G) dorsal view; (H) right lateral view; (I) red-cyan stereo anaglyph, anterolateral view; (J) anterior view; (K) ventral view, presumed pyrite crystals in red; (L) right lateral, presumed pyrite crystals in red.

punctate. Based on its overall shape and distinct punctation it has strong similarities to species of the non-marine group *Harbinia* *Tsao, 1959* (an ingroup of Cypridoidea). Syninclusions: cf. Psychodidae (Diptera); *Alavesia* sp. (Diptera); Coleoptera sp.; Euarthropoda sp. (one unidentifiable remain and three isolated legs); Cymothoida sp. (Isopoda) (*Schädel, Hyžný & Haug, in press*).

## DISCUSSION

Even though the studied ostracod specimens are well-preserved, their positions within the amber piece do not allow us to study further details, so we are unable to confidently identify them beyond being representatives of Cypridocopina. This limitation in turn impedes the palaeoenviromental interpretation of the assemblage. Nevertheless, we can speculate, on the basis of gross morphological similarities, that at least taxa A and B belong to either of two ingroups of Cypridocopina, namely Pontocypridoidea and Cypridoidea. Moreover, as we don't know if the specimens are adults, and the shape of Taxon A could be a juvenile, which tend to have less inflated posterior margins. If we assume Taxon A is an adult, then it resembles species of Pontocypridoidea or Cypridoidea (perhaps Paracypridinae, an ingroup

of Candonidae). Cypridoidea is the predominant group in non-marine environments today, such as fresh water and saline inland water (athalassic environments). Pontocypridoidea, on the other hand, comprises marine and brackish-water taxa (*Horne, 2003*). In the case of Taxon C the general shape and the association with the other two taxa suggest that it, too, is a cypridocopine. Species of the Cretaceous non-marine cypridoidean genus *Harbinia Tsao, 1959*, have similar carapace shape and punctate/reticulate ornament (e.g., *Harbinia hapla Tsao, 1959*, illustrated by *Ye et al. (2003*: pl. 27, figs 2a-c). However, we cannot rule out the possibility that it may belong to the group Cytheroidea (which is the dominant marine group today but also has non-marine lineages) without being able to see the characteristic adductor muscle scar patterns that allow to distinguish between cypridoideans and cytheroideans.

All three ostracod specimens in this study lack preserved soft parts but are preserved with articulated carapaces, suggesting limited (if any) post-mortem transport. In view of our uncertainty about the precise systematic interpretation of our specimens, we can contribute little to the discussion of the taphonomic and palaeoenvironmental interpretation of the Burmese amber assemblage. Both cypridoidean and pontocypridoidean taxa would be consistent with the mixed marine–freshwater–terrestrial components of the assemblage.

An ammonite shell preserved in Burmese amber argues that the Burmese amber forest was located near a dynamic and shifting coastal environment (*Yu et al., 2019*), a conclusion supported by the occurrence of a marine myodocopan ostracod (*Xing et al., 2018*). Also, semi-aquatic and aquatic insects occur in Burmese amber, including Ochteridae (Hemiptera), Heteroceridae (Coleoptera), Chresmododea and Gerridae (Hemiptera), Dytiscidae and Gyrinidae (Coleoptera), adults and larvae of Odonata, larvae of Psephenidae, Trichoptera, and Ephemeroptera (*Zhang, 2017*; *Xing et al., 2018*; *Schädel, Müller & Haug, 2020*). The herein presented ostracods are within the same amber piece as a fossil, supposedly aquatic living, isopod (*Schädel, Hyžný & Haug, in press*). Actuo-palaeontological experiments (*Schmidt & Dilcher, 2007*) have demonstrated that it is easily possible for aquatic organisms to be trapped in submerged bodies of resin. Several records of delicate arthropod remains from groups with supposed aquatic lifestyle (*Heard, De Lourdes Serrano-Sánchez & Vega, 2018*; *Schädel, Perrichot & Haug, 2019*; *Schädel, Müller & Haug, 2020*; *Serrano-Sánchez et al., 2015*; *Serrano-Sánchez et al., 2016*) indicate that the result of in-situ embedment of aquatic organisms is present in many amber sites. Recurrent flows of resin and changing water levels can explain the preservation of aquatic and non-aquatic organisms—such as the dipterans in the herein presented assemblage—in the same amber piece (*Xing et al., 2018*).

## CONCLUSIONS

The three specimens are new records of podocopan ostracods to be reported from Burmese Cretaceous amber, and most likely belong to the group Pododocopida, more precisely to its ingroup Cypridocopina. Within Cypridocopina the fossils either belong to the group Cypridoidea (indicative of non-marine environments) or to the group Pontocypridoidea (indicative of marine/brackish environments). If the former is true, these would be the

oldest non-marine ostracods preserved in amber. A position within either of the ingroups of Cypridocopina would be consistent with previous evidence of a mixed marine–freshwater–terrestrial assemblage deposited in a coastal setting.

## ACKNOWLEDGEMENTS

Our greatest gratitude is to Mark Pankowski who generously donated the amber piece to the Natural History Museum in Vienna. We thank Matúš Hyžný and Mathias Harzhauser for mediation and access to the amber piece. We are grateful to Roland Melzer (Zoological State Collection, Munich) for his help with the CT scans. Viktor Baranov (LMU, Munich) is thanked for his help with the identification of the syninclusions. MS thanks Joachim T. Haug and Carolin Haug (both LMU, Munich) for their enormous support. For constructive suggestions regarding the manuscript we thank the editor (Kenneth De Baets), Alan Lord and one anonymous reviewer. Furthermore, we are thankful to the various contributors of the free and open source software used in this study.

### Funding

This work was supported by Austrian Science Fund (FWF) project P 27687-N29, the Deutsche Forschungsgemeinschaft (DFG Ha 6300/3-2) and the Strategic Priority Research Program of the Chinese Academy of Sciences (XDB26000000). The funders had no role in study design, data collection and analysis, decision to publish, or preparation of the manuscript.

### Grant Disclosures

The following grant information was disclosed by the authors:
Austrian Science Fund (FWF): 27687-N29.
Deutsche Forschungsgemeinschaft: DFG Ha 6300/3-2.
Strategic Priority Research Program of the Chinese Academy of Sciences: XDB26000000.

### Competing Interests

The authors declare there are no competing interests.

### Author Contributions

- He Wang conceived and designed the experiments, prepared figures and/or tables, authored or reviewed drafts of the paper, and approved the final draft.
- Mario Schädel conceived and designed the experiments, performed the experiments, prepared figures and/or tables, authored or reviewed drafts of the paper, and approved the final draft.
- Benjamin Sames and David J. Horne analyzed the data, authored or reviewed drafts of the paper, and approved the final draft.
## Data Availability

The provenance of the Burmese amber meets the Society of Vertebrate Paleontology guidelines (April 21, 2020).

The amber is deposited permanently at the Natural History Museum Vienna (Naturhistorisches Museum Wien', NHMW) in full compliance with the International Code of Zoological Nomenclature (https://www.iczn.org/the-code/the-international-code-of-zoological-nomenclature/the-code-online/). The fossils at the NHMW are held safely in trust for the benefit of researchers and educators in the world respecting all ethnic groups, ages, sexes, landowners and collectors. Apart from public exhibitions, access is free to all scientists and interested people by prior arrangement during normal working hours and subject to the NHMW laboratory and museum regulations.

All data related to the paper are also available in the Morph.D.Base (Zoological Research Museum Alexander Koenig, Germany): M_Schaedel_20200706-M-34.1, https://www.morphdbase.de/?M_Schaedel_20200706-M-34.1.

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
