# Peer review of "New record of podocopid ostracods from Cretaceous amber"

_PeerJ, doi:10.7717/peerj.10134_

## Round 0.1 · original submission · Minor Revisions

This is a well written manuscript documenting the first podocopan ostracods in Cretaceous Burmese Amber, which I would like to see published. There are just some minor, but crucial points which need to be resolved before publication. The main points are:

Dates of Purchase and Donation: Please provide the provenance and dates (as precisely as possible) when the specimen was purchased and when it was donated to the Museum. See also comments by Reviewer 1. This might seem particularly relevant for Burmese amber but should be standard good practice for all fossils.

Taxonomic uncertainty: you mention that the 3 specimens belong to 3 different species, but I would like to see more explicit information on why these specimens likely belong to different species (which traits speak for this, how do specimens/species differ from one another). Please provide measurements for all specimens (compare comments by Reviewer 1). Are you certain these specimens are adult (see also comment by reviewer 1)? It is a bit confusing as you only compare A and C explicitly with taxa in Cypridoidea in the taxonomic description in the taxonomic description, while you compare A and B in the discussion. Pontocypridoidea is not mention at all in the taxonomic description. Also, in the case of B - not explicit comparisons are made or similarities are mentioned in the taxonomic description. Please resolve these inconsistencies.

Taxonomic discrepancies: you are rightfully cautious in your interpretations give the lack of certain traits (compare also with comments by reviewer 1, 2), but there are some inconsistencies between taxonomic assignment/comparison mentioned in the species description and the discussion.

Paleoenvironmental/paleoecological conclusions: please take this opportunity to discuss in a bit more detail the possible constraints given by the preservation (tightly closed valves), possible affinities (and their relationship with swimming abilities) and the associated syninclusions on the environmental to depositional conditions of the amber piece (see particularly the comments by reviewer 1)

Figures: there are also some inconsistencies (e.g., left versus right valve enlargement, presence of a shallow indentation not mention in text) between the descriptions and the figures (see comments by both reviewer 1 and 2).

Missing Reference: Please cite missing references and primary references rather than review paper where appropriate (see comments by reviewer 1).

Please make sure to address these and additional comments/suggestions by the reviewers and annotated pdfs.

Reviewer 1 ·

Basic reporting

The article is clear, concise and well structured. The English is generally of a reasonable standard, but sounds a bit awkward here and there, e.g. line 24, 29, 39 etc. The last author should be able to sort out the English.

The literature references are mostly fine, apart from the following;
I suggest that original sources of data for specific facts should be cited rather than general review chapters on ostracods. For example (line 28–39),
“…the marine Gigantocypris can be up to 32 mm (Horne et al., 2002; Brusca & Brusca, 2003).”
The original data for the size of Gigantocypris probably came from Poulsen 1962, who recorded a maximum size of 34 mm for Gigantocypris agassizi (credit where credit is due).
Poulsen, E. M. 1962. Ostracoda-Myodocopa, 1: Cypridiformes-Cypridinidae. In Dana Report, 57:1-414, 181 figures. Copenhagen: Carlsberg Foundation.

One reference is missing: Line 47–48.
“However, most of the relatively few records of ostracods preserved in amber (e.g., Keyser & Friedrich, 2017; Matzke-Karasz et al., 2019)…”
There is also Keyser, D. & Weitschat, W. 2005. First record of ostracods (Crustacea) in Baltic amber. Hydrobiologia, 538, 107–114.

The references need checking through. Smith et al. 2011 is not cited in the text. Smith et al. 2015 (line 82) is not listed in the references. I'm not sure either Smith paper is an appropriate citation for the higher classification of ostracods.

The figures are of good quality, but there is a mistake on the scale bar of Figure 1A. Additionally, the red-cyan stereo anaglyphs (Fig. 2, D and I) didn’t work for me. A photograph of the whole amber piece would also be a good idea.

Experimental design

With the ethical controversy surrounding Burmese amber, further details about how this piece was obtained should be given, such as its provenance, date of purchase by previous owners, and date of donation/purchase by the Natural History Museum of Vienna. Whether this is an ethically sourced specimen needs to ascertained before this paper is published. The Journal of Systematic Palaeontology is now refusing to publish any articles on Burmese amber, but of course, this is a matter for the editor.

Validity of the findings

This is an interesting article about (possible non-marine) ostracods from Burmese amber. The authors have been quite conservative in their conclusions about what the ostracods can tell us and which taxa they represent, which is a good thing. I agree with their assessment, although I think they have missed some possible ‘clues’ about these ostracods (see below).

There does seem to be a small discrepancy about possible taxonomic affinities in the article. The Remarks section of Taxon A concludes “… perhaps favours an assignment to the group Cypridoidea but this is not conclusive.” (line 122).
However, the Discussion (line 170) says:
“Nevertheless, we can speculate, on the basis of gross morphological similarities, that at least taxa A and B belong to either of two ingroups of Cypridocopina, namely Pontocypridoidea and Cypridoidea.”

One thing I would mention in the Discussion is that we don’t know if the specimens are adults. They are all quite small, and the shape of Taxon A could be a juvenile, which tend to have less inflated posterior margins.
If we assume Taxon A is an adult, then it resembles a Pontocypridoidea or Cypridoidea (maybe even a Paracypridinae of the Candonidae: Cypridoidea).

The ostracods were living in the same environment as the isopod, so does the isopod give any clues to the palaeo-environment? (e.g. a marine, brackish or freshwater taxon?). The main problem with Burmese amber is the lack of stratigraphic data with the pieces recovered. Some think that Burmese amber may span at least 5 million years (according to Science, 24 May 2019 issue, p. 725). So it is unknown if this piece came from the same time period as the ammonite or the marine myodocopid ostracod. Therefore, previous finds may not be that informative (or even misleading) with respect to the palaeo-environment of isolated amber pieces. I would therefore place more weight on syninclusions; the Diptera and Coleoptera would suggest non-marine for this piece, which in turn nudges the possible taxonomic affinities of the ostracods towards the Cypridoidea.

The fact that the ostracods are totally surrounded by amber, with no evidence of a substrate near them, suggests that they were capable of swimming above the substrate and they ‘landed’ on the resin. (It also suggests that the ostracods were alive when trapped as it would not be possible for dead ostracods on a substrate to be cleanly separated out like this.) This would exclude the Cytheroidea, and Candoninae (can’t swim). Could the ostracods have been feeding on, or drawn to, a trapped isopod, and consequently been trapped themselves? I think that one study of all the inclusions in this piece, rather than being split in different publications, would have been a better approach to understanding the palaeo-environment. But ultimately that is up to the authors.

Other records of ostracods in amber typically have a gape to the valves and appendages protruding. These ostracds don’t, but are instead tightly closed. I don’t know the significance of this, but perhaps it should be noted somewhere.

Additional comments

Other small issues with the manuscript style include:

Line 22
The first line of the abstract is a bit limp, and could be changed to something like:
“Burmese Cretaceous amber (~99 Ma, Myanmar) is famous for the preservation of a wide range of fauna and flora, including representatives of marine, freshwater and terrestrial groups.

Line 30
“Taphonomic and palaeoenvironmental implications are also discussed.”
Rather than just saying they are discussed, the authors should outline their conclusions.

I suggest adding that tomography was used to study the ostracods to the abstract.

Line 39.
“Ostracods are also the most common fossil arthropods during the geological history of the group, which can be dated back to Early Ordovician (Horne, 2005).”
A clearer way to say this is perhaps “Ostracods are the most commonly preserved fossil arthropods…”.

Line 61.
What is the size of the amber piece?

Line 97.
“Fig. 1A; Fig. 2; Suppl. 1”
Should note that it is marked by the arrow in fig.1A (not the whole plate).

Line 101
There are dimensions for Taxon A, but not for B and C. Is it possible to get a e.g. length, if not all dimensions for the other two?

Line 103
“Locality and horizon.”
Add latitude and longitude (also for the other two taxa).

Line 118
“Remarks: The right valve shows shallow indentation near the anterior end of the ventral margin…”
Surely you mean the left valve, as seen in Fig. 2H?
An arrow on the figure, and the figure number would help here. This shallow indentation is not mentioned in the description. OK, so you think it is possibly an artefact, but I think it should be included.

Line 160
“Alavesia sp. (Dipera);”
Should be Diptera.

Line 177
“However, we cannot rule out the possibility that it may belong to the group Cytheroidea…”
Unlikely to be Cytheroidea because they don’t swim. See comment under Validity of the findings.

Line 204
Better to write
“The three specimens are the first podocopan ostracods to be reported from Burmese Cretaceous amber”

·

Basic reporting

The manuscript reports three ostracods preserved in Cretaceous age amber from Burma. Relatively few ostracods have been described from amber and similar deposits and, as the authors observe, more from Cenozoic age deposits than from Mesozoic. The material is of international interest, although that I predict in a few years time interest will focus only on ostracod specimens with soft parts preserved.

Experimental design

Aims simple - to document a currently rather rare record.
Methods clearly described and appropriate.

The manuscript is concise, suitably structured, generally well written, the experimental design and methodology are appropriate, reference citations are relevant and up-to-date, and the figures are both necessary and of good quality.

Validity of the findings

See my comments (attached) on the identifications. Otherwise data justify conclusions.

Additional comments

This is a rather simple piece of science but, as I note in my attached comments, of current international interest.

---

## Round 0.2 · Minor Revisions

Thank for revising your manuscript and providing more details on the provenance of the specimen. Please considering citing Haug et al. (2020; https://doi.org/10.1007/s12542-020-00524-9) for readers not familiar with the background situation. Your manuscript is as good as accepted. There are just some minor additional points i noticed in the revised manuscript i would like to address/consider before paper: see annotated pdf.

---

## Round 0.3 · accepted · Accept

Thank you for integrating my final suggestions.